# Ecological drivers of CRISPR immune systems

Wei Xiao,[1] J. L. Weissman,[2,3,4] Philip L. F. Johnson[1]

**ABSTRACT** CRISPR-Cas is the only known adaptive immune system of prokaryotes. It is a powerful defense system against mobile genetic elements such as bacteriophages. While CRISPR-Cas systems can be found throughout the prokaryotic tree of life, they are distributed unevenly across taxa and environments. Since adaptive immunity is more useful in environments where pathogens persist or reoccur, the density and/or diversity of the host/pathogen community may drive the uneven distribution of CRISPR systems. We directly tested hypotheses connecting CRISPR incidence with prokaryotic density/diversity by analyzing 16S rRNA and metagenomic data from publicly available environmental sequencing projects. In terms of density, we found that CRISPR systems are significantly favored in lower abundance (less dense) taxa and disfavored in higher abundance taxa, at least in marine environments. When we extended this work to compare taxonomic diversity between samples, we found CRISPR system incidence strongly correlated with diversity in human oral environments. Together, these observations confirm that, at least in certain types of environments, the prokaryotic ecological context indeed plays a key role in selecting for CRISPR immunity.

**IMPORTANCE** Microbes must constantly defend themselves against viral pathogens, and a large proportion of prokaryotes do so using the highly effective CRISPR-Cas adaptive immune system. However, many prokaryotes do not. We investigated the ecological factors behind this uneven distribution of CRISPR-Cas immune systems in natural microbial populations. We found strong patterns linking CRISPR-Cas systems to prokaryotic density within ocean environments and to prokaryotic diversity within human oral environments. Our study validates previous within-lab experimental results that suggested these factors might be important and confirms that local environment and ecological context interact to select for CRISPR immunity.

**KEYWORDS** CRISPR, prokaryotes, adaptive immunity, evolution

Bacteria and archaea live constantly under the threat of viral infection (1, 2). In response, they have evolved an arsenal of defense systems of which the CRISPR (clustered regularly interspaced short palindromic repeats)-Cas (CRISPR-associated genes) system is the only known microbial adaptive immune system. As a testament to its efficacy, CRISPR can be found throughout the prokaryotic tree of life. However, the distribution of CRISPR is uneven among taxa and environments. CRISPR presents in most (~90%) sequenced bacterial thermophiles and archaea (both mesophilic and thermophilic) but in less than half (~40%) of mesophilic bacteria (3, 4). Metagenomic sequencing of environmental samples has also revealed uncultured bacterial lineages that lack CRISPR (5) despite its presence in closely related lineages. Furthermore, CRISPR loci can move into new lineages via horizontal gene transfer (6, 7). This ubiquity of opportunity to acquire a CRISPR-Cas system raises a question: what ecological factors drive the uneven CRISPR distribution?

CRISPR and other immune systems experience selection as prokaryotes adapt to their pathogen environment (8). Studies have shown that CRISPR-Cas-mediated immunity

Address correspondence to Philip L. F. Johnson, plfj@umd.edu.

The authors declare no conflict of interest.

See the funding table on p. 10.

in bacteria comes with inducible costs upon phage infection (9, 10). These costs arise directly from the expression of the CRISPR array and *cas* genes upon infection (11, 12) as well as indirectly from the time lag between the start of phage gene expression and clearance by the CRISPR system (13, 14). Even short-term expression of phage genes can be toxic and leads to reduced fitness of the host (13). Experimental work in bacteria has shown that a gradual increase in phage exposure selects for decreased levels of CRISPR-Cas-mediated immunity relative to surface modification (sm)-mediated immunity (10, 15), which would imply a negative association between phage quantity and CRISPR immunity. Consistent with these results, theoretical work has predicted selection should favor CRISPR-like adaptive immunity in a low-pathogen environment (16). In contrast, a recent metagenomic analysis of diverse natural environments revealed a positive correlation between viral density and CRISPR prevalence (17). These discrepant results may be due to the difference in biological complexity between the controlled experimental/theoretical conditions and natural environments. Regardless, both observations suggest that the viral abundance is involved in shaping the CRISPR system distribution.

In addition to viral population properties, prokaryotic population properties also affect the CRISPR distribution. An experiment that focused on bacterial biodiversity found the host was more likely to acquire CRISPR-based resistance when co-cultured with a mixed community containing two to four bacterial species than when cultured axenically (18). The authors hypothesize that developing alternative sm-mediated immunity may sacrifice other functions of cell surface molecules crucial for competing with other species for metabolic resources. Even in the absence of competing species, nutrient concentrations directly affect bacteria density with experiments showing lower nutrient concentrations favor a CRISPR-mediated inducible defense system over a sm-mediated constitutive defense system (10). Furthermore, the ecological and evolutionary interplay between host and virus means that both prokaryotic diversity and density will be indirectly linked to CRISPR immune strategy distribution via their association with viral density (19–21). Given that most phages are only capable of infecting a narrow range of host bacteria (22), the density of a bacterial host generally correlates with the density of its phage pathogens (23).

In this paper, we test these hypothesized connections between bacterial density/diversity and CRISPR incidence in natural populations rather than previous approaches of simplified experimental systems or theoretical models. We use data from direct environmental sampling via 16S rRNA and whole-genome shotgun metagenomic sequencing, and we annotate CRISPR presence both using metagenomic sequencing data as well as complete genome sequences from GenBank. We found strong associations with both bacterial density and diversity, although the signal appears environment specific, which suggests that other ecological factors play a role as well.

## MATERIALS AND METHODS

### CRISPR annotation

The CRISPRCas database (CRISPRCasdb) contains annotations of CRISPR arrays and *cas* genes for all complete genomes in GenBank (24). CRISPR annotation was performed by the CRISPRCasFinder algorithm (25). We downloaded the contents of this database from https://crisprcas.i2bc.paris-saclay.fr/ as of August 2021. We consider CRISPR-containing strains to be those with an evidence level 4 array. Given this strain-level CRISPR annotation, we define species-level CRISPR incidence as the proportion of CRISPR-containing strains of the species. Moving to a higher taxonomic level, we define genus-level CRISPR incidence as the mean of the species-level CRISPR incidence. In this way, we obtained CRISPR incidence for 7,167 unique species and 1,559 unique genera.

Separate from the pre-existing annotations of complete isolate genomes, we generated new CRISPR annotations from 2,631 draft Tara Oceans metagenomic-assembled genomes (MAGs) (26) with the CRISPRCasFinder program (25). We used default

settings to scan for the presence of CRISPR arrays and considered CRISPR-containing MAGs to be those with at least one evidence level 4 CRISPR array.

Finally, we annotated and quantified CRISPR from 2,355 human microbiome metagenomic samples (27). Human metagenomic shotgun sequence assemblies were obtained from Human Microbiome Project (HMP) portal hmasm2 (https://www.hmpdacc.org/hmp/hmasm2/) in December 2021. Human metagenomic shotgun sequence raw reads were downloaded from HMP portal HMWGSQC2 (https://www.hmpdacc.org/hmp/hmwgsqc2/) in March 2022. Assemblies were scanned for the presence of CRISPR arrays using CRISPRCasFinder with default settings. We then calculated a sample-wise measure of CRISPR load via quantifying CRISPR direct repeat abundance in each sample. We first built a database consisting of all unique direct repeats from evidence level 4 arrays along with their reverse complement sequences. For each of the 2,355 metagenomic samples, we inferred its sample-wise CRISPR activity level using direct repeat abundance as a proxy: we counted the number of raw reads containing a perfect match to our direct repeat database and normalized this count by the total number of reads in the sample. This approach averages CRISPR activity across all strains in a sample, which means it may be strongly influenced by a small number of taxa ubiquitous across samples. To mitigate this potential influence, we clustered repeats based on sequence similarities using CD-HIT (28) with an 80% sequence identity threshold (around 6 bp difference in repeat sequences) and a word size of 5 (cd-hit-est -c 0.8 n 5) and then excluded read counts mapping to repeat clusters present in more than 50% of the samples. We used the read counts that map to the remaining repeats as our final estimate for sample-wise CRISPR activity.

## Taxon abundance calculation

### Tara Oceans 16S rRNA data

The Tara Oceans team systematically collected samples from three ocean layers (surface water layer [SRF], deep chlorophyll maximum layer [DCM], and mesopelagic zone [MES]) around the world (29). Our analysis includes all 135 prokaryote-enriched samples of the 3 ocean layers. The team sequenced and clustered the 16S ribosomal RNA genes in each sample into species-level operational taxonomic units (OTUs). We downloaded the OTU count table from the Tara Oceans Project data portal (http://ocean-microbiome.embl.de/companion.html) in January 2019. The OTU abundance was grouped at the genus level to reduce the extent of missing data when we later analyzed in conjunction with CRISPR incidence. The absolute abundance of each genus was obtained by taking the product of the relative abundance and cell counts of each sample (flow cytometer data were downloaded from the Tara Oceans Project data portal).

### Earth Microbiome Project 16S rRNA data

The Earth Microbiome Project (EMP) is composed of a great number of samples from independent studies that span diverse environments. Sample processing, sequencing, and core amplicon data analysis were performed with a standardized EMP protocol (30). We downloaded the OTU count table from the EMP data portal (http://ftp.microbio.me/emp/release1/) in August 2019 and grouped OTU abundance at the genus level. Our analysis includes all 684 samples of saline water environments.

### Tara Oceans MAGs

We obtained published length-normalized estimation of relative abundance of MAGs (26). The relative abundance of a MAG is defined as the ratio between read counts that map to single-copy marker genes of the MAG and the read counts that map to a set of markers that can be used to approximate the total bacterial and archaeal community of the sample.

## Diversity calculation

### Human Microbiome Project 16S rRNA data

The HMP used 16S rRNA sequencing to characterize the complexity of the human microbiome at five body sites by clustering into OTUs (31, 32). We downloaded the OTU count table from the NIH HMP data browser (https://www.hmpdacc.org/hmp/resources/data_browser.php) in April 2019. We limited our analysis to the ~85% of samples with ≥2,000 reads and downsampled to exactly 2,000 reads. We grouped reads at the genus level in each sample to maximize overlap with CRISPR annotations from CRISPRCasdb. We summarized the $\alpha$-diversity of each sample using Shannon's index (33) $H = -\sum_i p_i \ln p_i$, where $p_i$ is the relative frequency of genus $i$. For all pairs of samples $i$ and $j$ with similar Shannon's diversity [$|H_i - H_j|/ \max(H_i, H_j) \leq 0.05$], we also quantified $\beta$-diversity with Bray-Curtis dissimilarity [$1 - 2C_{ij}/(S_i + S_j)$], where $C_{ij}$ is the sum of the lesser counts of genera present in both samples, and $S_i$ and $S_j$ are the total counts of all genera in each sample.

### EMP 16S rRNA data

The EMP study #1774 (Puerto Rico and Plantanal samples for the western acculturation project) collected samples from different human body sites. All of its oral samples ($n = 121$) were included in our study. We obtained the Shannon's diversity index of each sample from the work of the EMP group (http://ftp.microbio.me/emp/release1/) in August 2019 (30).

### HMP metagenomic data

HMP provides 2,355 metagenomic samples (comprising both HMP phase I and phase II) targeting diverse body sites with multiple time points in 265 individuals (27). Shannon's diversity index was calculated at the species level from taxonomic profiling of metagenomic reads as reported by the NIH HMP data browser (https://www.hmpdacc.org/hmsmcp2/) in February 2020.

## RESULTS

### CRISPR is favored in lower abundance genera in marine environments

We started our investigation of CRISPR and prokaryotic density by analyzing 16S rRNA metagenomic data from the Tara Oceans Project (29) samples collected over the globe and using CRISPR annotations from the CRISPRCasdb (24).

For each of the 135 Tara Oceans samples, we sorted genera based on their relative abundance within the sample, divided this list of genera in half, and compared the abundance-weighted average CRISPR incidence of the less abundant genera to the more abundant genera. We found a clear signal with almost every sample having a higher CRISPR incidence among less abundant genera than more abundant genera (Fig. 1A; Data Set S1). This signal persists when restricted to only archaeal genera or only bacterial genera (Fig. S8; Data Set S1). This result aligns with the predictions from both theoretical models (16) and laboratory studies (10, 15) that suggest that lower density microbes also have lower density of viral pathogens, which in turn may provide a selective advantage for CRISPR-style immunity over surface immunity (4, 34). That said, other analyses of metagenomic data (17) have found a positive trend between viral density and CRISPR incidence, which would suggest lower viral density has a comparative disadvantage for CRISPR. Resolving this apparent discrepancy would ideally require inferring viral density directly from our data using viral-specific sequence abundances, but identifying such sequences is inherently unreliable with false negatives due to the rapid pace of viral genome evolution and false positives due to viral sequences becoming incorporated into prokaryotic genomes (35).

We next developed a more refined test to allow accounting for the possibility that this result is driven by a small number of highly abundant genera that occur in a large

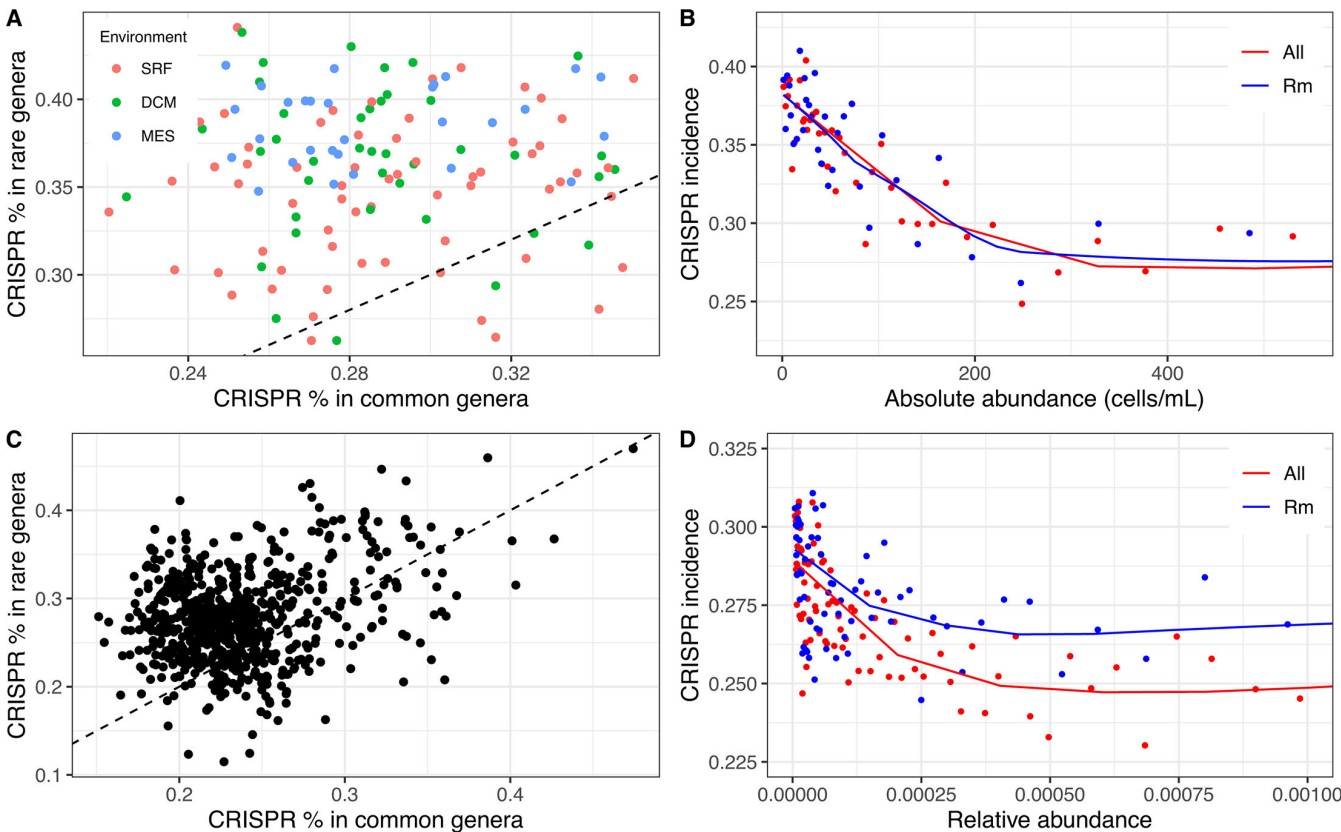

**FIG 1** Lower density associates with CRISPR incidence. Density analysis was performed with 16S rRNA data. (A) Within a sample, CRISPR is found at a higher rate in rare genera (bottom 50% of abundances) than common genera (top 50% of abundances). Each point represents a single sample from the Tara Oceans Project. Points are colored by different sampling depth. SRF, 5 m; DCM, 71 m; MES, 600 m. (B) The negative correlation between CRISPR incidence and absolute abundance was driven by both prevalent and rare genera. Each point represents the median CRISPR incidence of 500 genera with a similar absolute abundance of the Tara Oceans Project. Red points ("All") are summarized data with prevalent + rare genera, while blue points ("Rm") are summarized data after removing prevalent genera (present >75% of samples). For clearer trends visualization, we focus only on the lower 80% of absolute abundance (see Fig. S7 for the whole abundance range). (C, D) Similar analysis as in panels A and B but using saline water samples from the Earth Microbiome Project revealed a similar pattern. Flow cytometry data were not available for the EMP data, so panel D shows relative density instead of absolute density.

proportion of samples, which we define as "prevalent genera." We sorted the absolute abundance of genera in all samples together and smoothed the data by taking the median of the CRISPR incidence in bins of 500 abundance observations (Fig. 1B, red line; Data Set S1). This test replicated our initial finding of a negative relationship between CRISPR incidence and the absolute abundance and confirmed that it holds across a wide range of abundances. We then ruled out a small number of prevalent genera as the sole driver of this pattern by removing all genera present in more than 75% of samples and observing that the pattern persists (Fig. 1B, blue line). Some of these removed prevalent genera are also highly abundant such as *Pelagibacter*, *Prochlorococcus*, and *Synechococcus* (Data Set S2).

To further validate our results, we analyzed independent data from the Earth Microbiome Project and found a similar trend in samples from saline water environments (Fig. 1C and D; Data Set S3). Note the slight vertical shift between the two lines in Fig. 1D implies that the removed prevalent genera have lower average CRISPR incidence.

A caveat to the above analyses is that, by using CRISPR incidence from fully assembled prokaryotic genomes in the GenBank database (i.e., CRISPRCasdb), we are assuming that the CRISPR incidence in each genus is constant regardless of local environmental factors. We relaxed this assumption by annotating CRISPR directly in the environment in which we also assess abundance. More specifically, we searched for CRISPR arrays in 2,631 draft Tara Oceans metagenome-assembled genomes (26) using CRISPRCasFinder

(25). The absolute abundance of MAGs with a CRISPR system was compared to that of MAGs without CRISPR in all three environments (Fig. 2; Data Set S4). In SRF and DCM, MAGs without the CRISPR system tend to be more abundant in samples, which is consistent with our previous 16S rRNA results. The pattern is, in general, consistent among different geographic provinces (Fig. S6). The deepest layer (MES) does not show this pattern.

## CRISPR incidence increases with sample diversity in the human microbiome

The majority of prokaryotes on earth occur in biofilms (36), which are usually composed of multiple species (37). In such situations, species constantly undergo synergistic or antagonistic cross-species interactions (38), which may affect the relative fitness of different immune systems. We sought to empirically test for a link between environmental prokaryotic diversity and CRISPR immunity. We first examined the Tara Oceans data, where we found a significant positive correlation ($P$-value = 1.39e−07) in the SRF samples but no significant results ($P$-value = 0.056 for DCM and $P$-value = 0.018 for MES) in the other environments (Fig. S5; Data Set S1). Hypothesizing that the range of sample diversity would be a key factor in determining power, we decided to examine the human host-associated microbiome, which includes samples with significantly less diversity than Tara Oceans samples.

The Human Microbiome Project provides 16S rRNA sequencing data sampled from five human body sites (oral, nasal, skin, gut, and urogenital). We began by restricting ourselves to oral samples (which comprise half of the samples) and calculated Shannon's diversity index for each of the 2,564 oral samples using genus-level read counts. As in our initial density analysis, we used genus-level CRISPR annotations calculated from CRISPRCasdb. We observed a positive correlation with diversity explaining an amazingly high ~52.4% of the variation in CRISPR incidence (Fig. 3A; Data Set S5). This pattern

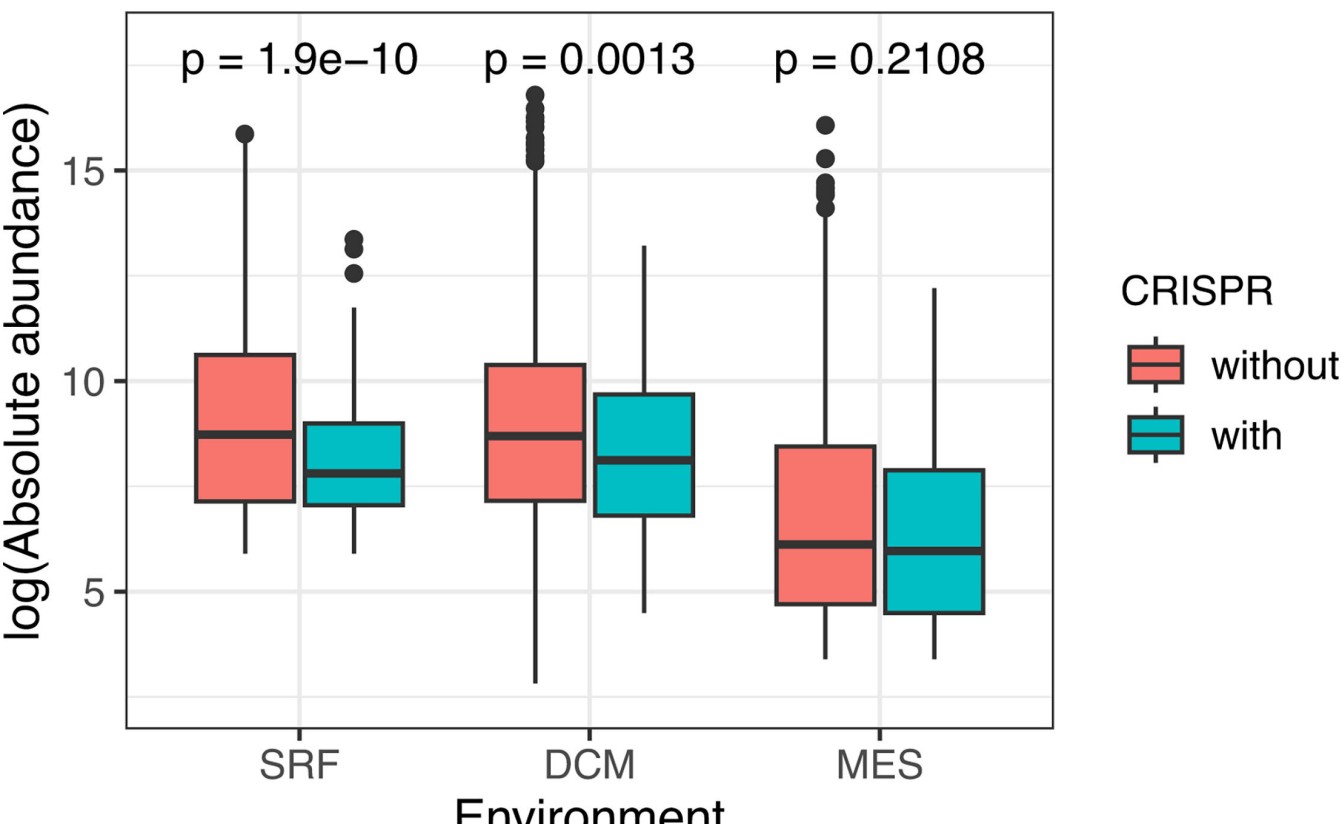

**FIG 2** At DCM and SRF depths in Tara Oceans data, MAGs without CRISPR array(s) are significantly more abundant than MAGs with CRISPR array(s).

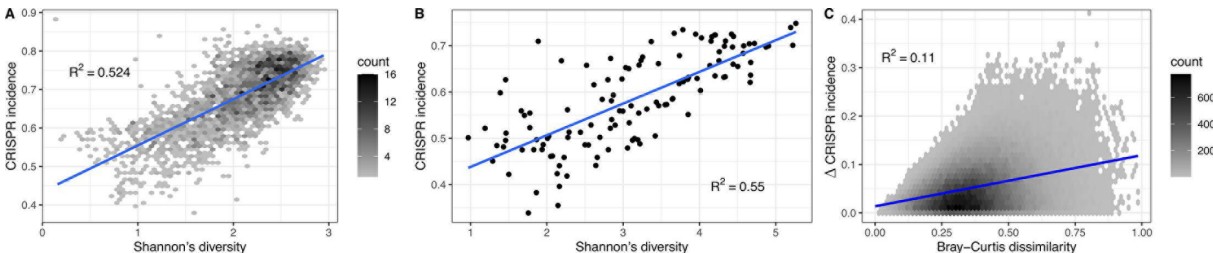

**FIG 3** CRISPR system incidence significantly correlates with microbial diversity in human oral environments. (A) Shannon's diversity index for 2,564 HMP oral samples was calculated with 16S rRNA sequencing data grouped at the genus level. CRISPR incidence came from CRISPRCasdb. (B) Same analysis with data from 121 human oral samples from EMP. (C) Difference in CRISPR incidence between pairs of HMP oral 16S samples with similar $\alpha$-diversity (Shannon's index within 5% of each other) positively correlates with Bray-Curtis dissimilarity.

varies among body sites with significant positive correlations in stool ($r^2 = 0.429$) and nasal cavity ($r^2 = 0.063$), a significant negative correlation in skin ($r^2 = 0.092$, opposite direction), and no meaningful correlation in urogenital tract ($r^2 = 0.018$) (Table 1; Fig. S1; Data Set S5).

As with the density analysis, we validated our diversity results with an independent set of human oral samples from the EMP and found a similar strong positive correlation (Fig. 3B; Data Set S6).

The different patterns among body sites suggest that the CRISPR incidence may be driven by species composition in addition to diversity. We assessed the effect of differences in species composition by examining CRISPR incidence as a function of Bray-Curtis dissimilarity between pairs of samples. During this analysis of between-sample diversity (i.e., $\beta$-diversity), we controlled for within-sample diversity ($\alpha$-diversity) by restricting to pairs of samples with Shannon's diversity within 5% of each other. Consistent with our species composition prediction, the CRISPR incidence difference increased along with between-sample diversity in the oral environment (Fig. 3C; Data Set S5) as well as all other body sites (Fig. S3; Data Set S5).

As noted in our earlier abundance analysis, CRISPR annotated from CRISPRCasdb does not reflect environmental factors. Thus, we directly tested the diversity hypothesis in HMP whole-genome shotgun metagenomic data (27), in which we directly annotated CRISPR incidence (see Materials and Methods). We compared the normalized CRISPR repeat abundance of 1,141 human oral samples against Shannon's diversity index and observed a small but highly significant positive correlation ($r^2 = 0.048$; Fig. 4A; Data Set S7). Results from all body sites can be found in Table 1; Fig. S2. To check if the trend is driven primarily by a small number of high-abundance genera present in most samples, we performed the same analysis after removing all CRISPR repeat clusters (clusters of repeats based on sequence similarities) present in >50% of samples. Given that repeat sequences associate with CRISPR-Cas system types and are phylogenetically conserved to some extent (39), this procedure should preferentially exclude abundant genera. The remaining 1,139 oral samples again show a positive correlation between CRISPR incidence and sample diversity ($r^2 = 0.053$; Fig. 4B; Data Set S7).

**TABLE 1** The table shows the coefficient of determination between CRISPR incidence and Shannon's diversity for all HMP body sites[a]

| Body site | 16S rRNA | | | Metagenomic | | |
|---|---|---|---|---|---|---|
| | Trend | $r^2$ | *P*-value | Trend | $r^2$ | *P*-value |
| Oral | Positive | 0.524 | 0 | Positive | 0.0481 | 6.61e−14 |
| Gut | Positive | 0.429 | 6.20e−34 | Positive | 0.0687 | 8.95e−10 |
| Nasal | Positive | 0.063 | 2.04e−03 | – | – | – |
| Skin | Negative | 0.092 | 1.46e−14 | – | 0.0064 | 0.4661 |
| Urogenital | – | 0.018 | 0.012 | – | 0.0129 | 0.1388 |

[a]A dash means either trend is not significant or data are not available.

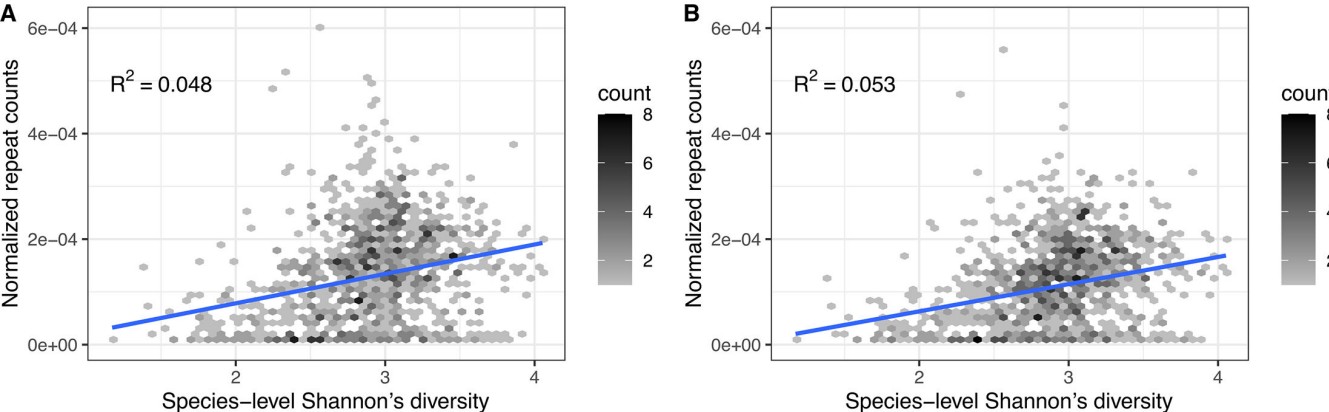

FIG 4 (A) Sample-wise repeat-mapped read counts positively correlated with Shannon's diversity index in 1,141 HMP human oral samples ($r^2 = 0.048$). (B) The positive correlation between repeat-mapped read counts and Shannon's diversity index in oral samples persists when prevalent repeat clusters (those present in more than 50% of samples) were removed ($r^2 = 0.053$).

## DISCUSSION

The CRISPR-Cas system is broadly found throughout the prokaryotic phylogeny and provides an effective immune response against viral infection. Defense genes generally have higher HGT rates (7), and *cas* genes and CRISPR arrays frequently physically cluster in defense "islands," which allow them to be horizontally transferred together (40). However, the CRISPR systems distribute unevenly across the prokaryotic phylogeny and environments (41) despite these high rates of HGT. This apparent paradox has been explained by direct fitness costs of the CRISPR system or reduced benefit due to competition from other immune systems (14, 18, 42–44). For instance, acquiring self-targeting spacers is deleterious to the host (45–47); CRISPR may prevent HGT so that the acquisition of advantageous mobile genetic elements is limited (48–51); CRISPR shows negative epistasis with host genes in some cases (41, 52). In this work, we explored additional ecological factors that correlated with CRISPR incidence and found that CRISPR is preferred in lower abundance taxa in saltwater environments as well as higher diversity samples in the human oral cavity.

### Density

Our analysis revealed that CRISPR is preferred in lower abundant prokaryotes in marine environments. One hypothesized reason could involve the classic kill-the-winner model of host-virus interaction in marine environments. Phages lyse ~20% of ocean bacteria every day (53, 54), and the most abundant bacteria are targeted by more diverse phages. For example, during Flavobacteria blooms in the ocean, many new flavophages (defined as phages targeting flavobacteria) have been identified (55). CRISPR is more efficient in environments with recurrent phage with minimal genetic changes, which is supported by a negative correlation between phage diversity and CRISPR incidence found in a metagenomic analysis (17). Thus, we expect CRISPR systems are more likely to be found in lower abundance prokaryotic species, which are targeted by less diverse viruses.

The lack of significant correlation in the MES layer in Fig. 2 could be due to different biological lifestyles in the different environments or due to technical biases in the MAG reconstruction method. In particular, the relative abundance of particle-attached bacteria increases in the deeper ocean, while free-living bacteria dominate the shallower layers (56). While density has a fairly straightforward interpretation for free-living bacteria, the fine-scale community spatial structure of particle-attached bacteria may render our measure of density irrelevant for microbe-phage dynamics in such an environment. In addition to this biological explanation, technical details of the MAG generation favor conserved genomes from the SRF and DCM layers at the expense

of the MES layer through a complex ascertainment process. Data from all layers in a geographic region were merged during assembly to maximize completeness (26) since the MAG quality depends strongly on the sequencing coverage. However, this merging process induces a bias in the MAGs toward the upper layers, which contain more cell counts than lower layers (Fig. 2). Subsequently, abundances are calculated in all layers for all MAGs, but these calculations can only be made for assembled genomes—hence the ascertainment bias. For taxa that primarily live in the upper layers, their reported abundance in the deepest layer is unlikely to be related to having CRISPR or not, which could explain the absence of a signal in our analysis. The converse is less of a factor because taxa that primarily live in the deepest layer are less likely to be assembled. Furthermore, MAGs will preferentially contain conserved core regions of a genome over more variable accessory regions due to their differential abundances in sub-strain-level populations. Consequently, the absence of CRISPR-Cas systems in some MAGs likely reflects a failure to capture this level of microdiversity (57, 58). We expect these biases of MAGs will be overcome in the future with the increasing depth of coverage in studies generating large numbers of single-cell-assembled genomes (59–61).

Surprisingly, when we performed a similar density analysis with 16S rRNA data from the Human Microbiome Project, we found no signal (Fig. S4). However, these two environments have radically different ecologies supporting distinct prokaryotic lifestyles. The upper layers of the ocean are relatively homogeneous environments with low absolute nutrient densities and correspondingly low bacterial densities; these bacteria primarily live a planktonic lifestyle (56). In contrast, host-associated bacteria in the human body encounter high absolute nutrient densities at a macroscale, but they live in highly heterogeneous structured environments; fine-scale structure can lead to variation in nutrient accessibility, which in turn can lead to variation in bacterial densities. For example, the human oral microbiome is constructed with surface-dependent bacteria, which tend to form biofilms with complex community spatial structures and a mixture of taxa. Distinctive niches within the oral cavity lead to distinct microbiome profiles (62). In this case, species density in each sample might be a poor indicator of the fine-scale population density relevant to host-virus dynamics.

## Diversity

In addition to the association with low abundance, we found CRISPR to be favored in human oral environments with higher prokaryotic diversity. This result confirmed previous experimental work directly showing that defense strategies such as sm-mediated immunity are selected against in more diverse bacteria communities (18). Many other factors potentially play a role in connecting CRISPR system distribution to prokaryotic diversity, although experimental evidence is less direct. First, studies found that microbial herd immunity is facilitated by spatially heterogeneous environments and community structures such as biofilms (63, 64) that correlate with diversity. Herd immunity requires CRISPR or CRISPR-like immunity in which the resistant strain acts as a phage sink to protect its neighbors. Second, in order to reduce the chances of phage evolving to escape CRISPR immunity, multiple spacers must target the same phage genome. While CRISPR arrays in an individual bacterium may have limited numbers of effective spacers (65, 66), spacer diversity at the population level provides an effective barrier to phage evolutionary escape (67). Such population-scale spacer diversity would occur naturally with greater diversity of host bacteria. Finally, in diverse but high-density microenvironments, such as biofilms, quorum sensing may drive population-level change in gene expression (68), including at least certain types of CRISPR immune responses (69). In the human oral cavity, bacteria present close interspecies synergistic interactions (70) in heterogenous microenvironments, which may link the community diversity with the CRISPR-associated properties of herd immunity, population-distributed immunity, and community-sensing immune plasticity.

When we extended this diversity analysis to the Tara Oceans samples, we found a mild trend in the SRF layer ($r^2 = 0.37$), but no correlation in the DCM and MES layers

(Fig. S5) was observed. However, compared to the human oral community, free-living prokaryotes that live in the ocean have unlimited space, with reduced opportunities for direct interaction and interspecies competition. The selective advantage of CRISPR relative to other immune mechanisms may thus be weaker, even though the average species richness of the ocean is several times higher than the oral cavity (29, 71).

Despite our ecological factors not being universal across environments, we revealed that prokaryotic community properties in natural environments correlate with the distribution of CRISPR systems. Indeed, the variation in trends suggests that nuanced changes in environmental conditions could result in substantial changes to selection on defense systems. These trends are likely also coupled to virus community properties. Future investigation of this hypothesis will require more extensive data sets with paired prokaryotic and viral sequences collected at the same location.

## ACKNOWLEDGMENTS

This research was supported in part by National Institutes of Health grant R21GM147759 to P.L.F.J.

## AUTHOR AFFILIATIONS

[1]Department of Biology, University of Maryland, College Park, Maryland, USA
[2]Department of Ecology and Evolution, Stony Brook University, Stony Brook, New York, USA
[3]Institute for Advanced Computational Science, Stony Brook University, Stony Brook, New York, USA
[4]Department of Biology, The City College of New York, New York, New York, USA

## AUTHOR ORCIDs

Wei Xiao  http://orcid.org/0000-0002-6313-3543
J. L. Weissman  http://orcid.org/0000-0002-4237-4807
Philip L. F. Johnson  http://orcid.org/0000-0001-6087-7064

## FUNDING

| Funder | Grant(s) | Author(s) |
| --- | --- | --- |
| HHS | National Institutes of Health (NIH) | R21GM147759 | Philip L. F. Johnson |

## AUTHOR CONTRIBUTIONS

Wei Xiao, Conceptualization, Data curation, Formal analysis, Investigation, Methodology, Software, Validation, Visualization, Writing – original draft, Writing – review and editing | J. L. Weissman, Conceptualization, Data curation, Methodology, Supervision, Writing – review and editing | Philip L. F. Johnson, Conceptualization, Data curation, Funding acquisition, Investigation, Methodology, Supervision, Writing – review and editing

## ADDITIONAL FILES

The following material is available online.

### Supplemental Material

**Supplemental figures (mSystems00568-24-s0001.pdf).** Figures S1 to S8.
**Legends (mSystems00568-24-s0002.pdf).** Supplemental data file legends.
**Dataset S1 (mSystems00568-24-s0003.xlsx).** Genus-level abundance and CRISPR incidence in each 16S rRNA sample of the Tara Oceans Project.
**Dataset S2 (mSystems00568-24-s0004.xlsx).** Summarized version of Data Set S1.

**Dataset S3 (mSystems00568-24-s0005.xlsx).** Genus-level abundance and CRISPR incidence in each saline-water 16S rRNA sample of the Earth Microbiome Project.

**Dataset S4 (mSystems00568-24-s0006.xlsx).** Abundance of each MAG and whether they encode CRISPR in each metagenomic sample of the Tara Oceans Project.

**Dataset S5 (mSystems00568-24-s0007.xlsx).** Genus-level abundance and CRISPR incidence in each 16S rRNA sample of the Human Microbiome Project.

**Dataset S6 (mSystems00568-24-s0008.xlsx).** Shannon diversity and CRISPR incidence of each 16S rRNA sample from human oral study #1774 in the Earth Microbiome Project.

**Dataset S7 (mSystems00568-24-s0009.xlsx).** Shannon diversity and CRISPR incidence of each metagenomic sample in the Human Microbiome Project.

## Open Peer Review

**PEER REVIEW HISTORY (review-history.pdf).** An accounting of the reviewer comments and feedback.

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
