## [Reviewer comments · mSystems]

Ecological drivers of CRISPR immune systems

Wei Xiao, JL Weissman, and Philip Johnson

Corresponding Author(s): Philip Johnson, University of Maryland at College Park

Review Timeline:

Submission Date:	May 8, 2024
Editorial Decision:	June 26, 2024
Revision Received:	August 28, 2024
Accepted:	September 26, 2024

Editor: Francisco Rodriguez-Valera

Reviewer(s): The reviewers have opted to remain anonymous.

Transaction Report:

DOI: <https://doi.org/10.1128/msystems.00568-24>

Re: mSystems00568-24 (Ecological drivers of CRISPR immune systems)

Dear Prof. Philip L F Johnson:

Revision Guidelines

Sincerely,
Francisco Rodriguez-Valera
Editor
mSystems

Reviewer #1 (Comments for the Author):

Xiao et al 2024 mSystems

Xiao et al present a very thorough study on the prevalence of CRISPR Cas systems in relation to prokaryotic diversity and density. These results greatly contributes the field of microbiology from both a biotechnological perspective, as diverse applications of CRISPR Cas systems are growing, as well as seeking new systems to work with. This study provides a starting

point, to some extent, of which environments CRISPR Cas systems are more likely found. Furthermore, in viral ecology, matching CRISPR spacers to viruses is a popular way to link viruses and hosts, and this study shows which environments this host-matching approach may or may not be as effective. The authors do not mention implications of their findings for these purposes, but their results are a great resource for such. Nonetheless, I have some concerns that need.

I have a few comments and concerns that impact methods and interpretation.

Firstly, this study lacks Supplemental Datasets with the results from the CRISPR Cas detection, among other information (e.g. the Shannon's diversity estimate for each sample). This prevents the reproducibility of this study. Although the authors describe which genomes, tools, and parameters were used, by not providing their results, someone who may replicate their work will not have anything to ensure they proceeded correctly. Additionally, this limits the use of its results - perhaps someone wants to know which Tara Oceans sample had the highest CRISPR Cas prevalence - there is no way to trace this from the manuscript as is). The authors should provide tables for each dataset examined with information on the genomes in each sample with CRISPR Cas detection results and other data used in results reported (e.g. Shannon's diversity estimates, Bray-curtis dissimilarities, etc.)

Regarding interpretation, another key component dictating the use of certain defense types is nutrient levels. One of the reference they mention [10] even demonstrates the shift from surface-receptor mutations to CRISPR-Cas mediated immunity as nutrient levels decrease. Nutrient levels is a major discrepancy in the two main environments the authors compare in this study (surface ocean (generally low nutrients) and human microbiome (generally higher nutrients than the surface ocean)). This is briefly mentioned when the authors find disparate trends in the Density discussion (lines 263-266) but should be emphasized earlier when other related observations are mentioned. I elaborate more specifically in my minor comments.

Regarding methods, some key limitations of the approaches are not acknowledged, namely in the use of MAGs (GenBank assemblies also include MAGs) to identify CRISPR Cas systems (particularly considering the co-assembly approach used for the Tara MAGs). The authors acknowledge that defense systems are highly mobile, [even phages can carry CRISPR Cas], but MAGs only assemble consensus genomes of populations, which can exclude defenses that vary within strains [Hussain et al 2022 Science]. Therefore, it is highly possible that the absence of CRISPR Cas systems in some MAGs/ MAGs in GenBank assessed may be missed since this level of microdiversity is missing from MAGs. These results should be more explicit that CRISPR Cas regions detected are those that are potentially conserved across populations within a strain rather than representing the true prevalence of CRISPR Cas in prokaryotes of that taxa in a given environment, given their emphasis on quantifying relationships.

Furthermore, to address this challenge of microdiversity complicating CRISPR Cas recovery and abundances, I recommend the authors also examine publicly available single-cell genome assembly (SAGs) datasets. SAGs capture individual-level diversity. Although SAGs are often more limited in genome completeness than MAGs, genome coverage is unbiased, and there are large datasets available that can account for this completion limitation to some degree; additionally, tools have even been developed to leverage both metagenomic and SAG data to enhance genome recovery [Arikawa et al 2021, Microbiome]. Furthermore, given that cells are unbiasedly sampled for sequencing in several SAG datasets, the authors could provide a more direct quantification of CRISPR Cas prevalence (in % of cells, for example). I am not suggesting the authors replace any results with SAG analyses but rather incorporate results from SAG datasets [Arikawa et al 2021 Microbiome, Munson McGee, Lindsay et al 2022 Nature] to provide more quantitative power to their findings, given that a key aim of this study to "empirically test for a link between environmental prokaryotic diversity and CRISPR immunity" [Lines 186-187].

Line 48-53: As mentioned in the general comments, this statement would be more meaningful if the authors acknowledged the impact of nutrient concentrations on the use of CRISPR Cas over surface receptor mutations (sm-mediated immunity). A key finding of reference 10 is that sm-mediated immunity is more common in high nutrient conditions, and low-nutrient conditions favor CRISPR Cas (likely due to cost of losing receptor in low nutrient environments). This would help explain discrepancies in diversity and density relationships with the prevalence of CC. This observation also ties into the results of the multi-species co-culture experiment mentioned in lines 55-59.

Line 166-167: Would be meaningful about the biology of these environments if examples of prevalent genera that were removed are mentioned e.g. *Pelagibacter*? *Prochlorococcus*? Knowing these groups would provide a lot more strength to their argument that highly abundant marine taxa almost never encode CC, as genomes from cultures of these organisms tend to lack CRISPR Cas.

Line 254-257: The authors should also state that using MAGs fail to recover highly abundant and diverse taxa (e.g. *Pelagibacter*; Roda-Garcia et al 2023, Environmental Micro.), which may skew their results.

Figure 1b,d: What is "All" and "Rm" - not clearly defined in text nor figure legend. I inferred "Rm" meant rare taxa but still unclear.

Reviewer #1 (Comments for the Author):

Xiao et al 2024 mSystems

Xiao et al present a very thorough study on the prevalence of CRISPR Cas systems in relation to prokaryotic diversity and density. These results greatly contribute to the field of microbiology from both a biotechnological perspective, as diverse applications of CRISPR Cas systems are growing, as well as seeking new systems to work with. This study provides a starting point, to some extent, of which environments CRISPR Cas systems are more likely found. Furthermore, in viral ecology, matching CRISPR spacers to viruses is a popular way to link viruses and hosts, and this study shows which environments this host-matching approach may or may not be as effective. The authors do not mention implications of their findings for these purposes, but their results are a great resource for such. Nonetheless, I have some concerns that need.

Thank you for your thorough review of our manuscript. We appreciate your positive feedback on the significance and contributions of our study to the field of microbiology, especially regarding the level of usefulness of CRISPR-Cas systems in linking host and virus in different environments.

I have a few comments and concerns that impact methods and interpretation.

To address your comments and concerns, we made the following modifications.

Firstly, this study lacks Supplemental Datasets with the results from the CRISPR Cas detection, among other information (e.g. the Shannon's diversity estimate for each sample). This prevents the reproducibility of this study. Although the authors describe which genomes, tools, and parameters were used, by not providing their results, someone who may replicate their work will not have anything to ensure they proceeded correctly. Additionally, this limits the use of its results - perhaps someone wants to know which Tara Oceans sample had the highest CRISPR Cas prevalence - there is no way to trace this from the manuscript as is). The authors should provide tables for each dataset examined with information on the genomes in each sample with CRISPR Cas detection results and other data used in results reported (e.g. Shannon's diversity estimates, Bray-curtis dissimilarities, etc.) .

We have added 7 supplementary datasets so that all figures and supplementary figures are easily reproducible. Here, we list brief descriptions of the 7 supplementary datasets.

- Supplementary dataset 1 includes the genera-level abundance and CRISPR incidence in each 16S rRNA sample of the Tara Oceans Project.
- Supplementary dataset 2 includes the average absolute abundance and prevalence count (the number of samples a genus can be found in) of each genus from the 16S rRNA samples in the Tara Oceans Project.
- Supplementary dataset 3 includes the genera-level abundance and CRISPR incidence in each saline-water environmental 16S rRNA sample of the Earth Microbiome Project.
- Supplementary dataset 4 includes the relative abundance of MAGs and whether they encode CRISPR or not in each metagenomic sample of the Tara Oceans Project.
- Supplementary dataset 5 includes the genera-level abundance and CRISPR incidence in each 16S rRNA sample of the Human Microbiome Project.
- Supplementary dataset 6 includes the Shannon diversity and CRISPR incidence of each 16S rRNA sample from the human oral study #1774 in the Earth Microbiome Project.
- Supplementary dataset 7 includes Shannon diversity and CRISPR incidence (represented by repeat-mapped read counts) of each metagenomic sample in the Human Microbiome Project.

Regarding interpretation, another key component dictating the use of certain defense types is nutrient levels. One of the reference they mention [10] even demonstrates the shift from surface-receptor mutations to CRISPR-Cas mediated immunity as nutrient levels decrease. Nutrient levels is a major discrepancy in the two main environments the authors compare in this study (surface ocean (generally low nutrients) and human microbiome (generally higher nutrients than the surface ocean)). This is briefly mentioned when the authors find disparate trends in the Density discussion (lines 263-266) but should be emphasized earlier when other related observations are mentioned. I elaborate more specifically in my minor comments.

The reviewer made a great point that we should emphasize the importance of nutrient levels in the introduction. We have added a consideration of how nutrient levels likely affect the linkage between prokaryotic density and immunity (lines 59-61).

Regarding methods, some key limitations of the approaches are not acknowledged, namely in the use of MAGs (GenBank assemblies also include MAGs) to identify CRISPR Cas systems (particularly considering the co-assembly approach used for the Tara MAGs). The authors acknowledge that defense systems are highly mobile, [even phages can carry CRISPR Cas], but MAGs only assemble consensus genomes of populations, which can exclude defenses that vary within strains [Hussain et al 2022 Science]. Therefore, it is highly possible that the absence

of CRISPR Cas systems in some MAGs/ MAGs in GenBank assessed may be missed since this level of microdiversity is missing from MAGs. These results should be more explicit that CRISPR Cas regions detected are those that are potentially conserved across populations within a strain rather than representing the true prevalence of CRISPR Cas in prokaryotes of that taxa in a given environment, given their emphasis on quantifying relationships.

Thanks to the reviewer for pointing out this limitation of MAGs. In our revision, we raise this possibility in the discussion section (lines 268-272).

Furthermore, to address this challenge of microdiversity complicating CRISPR Cas recovery and abundances, I recommend the authors also examine publicly available single-cell genome assembly (SAGs) datasets. SAGs capture individual-level diversity. Although SAGs are often more limited in genome completeness than MAGs, genome coverage is unbiased, and there are large datasets available that can account for this completion limitation to some degree; additionally, tools have even been developed to leverage both metagenomic and SAG data to enhance genome recovery [Arikawa et al 2021, Microbiome]. Furthermore, given that cells are unbiasedly sampled for sequencing in several SAG datasets, the authors could provide a more direct quantification of CRISPR Cas prevalence (in % of cells, for example). I am not suggesting the authors replace any results with SAG analyses but rather incorporate results from SAG datasets [Arikawa et al 2021 Microbiome, Munson McGee, Lindsay et al 2022 Nature] to provide more quantitative power to their findings, given that a key aim of this study to "empirically test for a link between environmental prokaryotic diversity and CRISPR immunity" [Lines 186-187].

The reviewer raises an excellent point that using SAGs would avoid the inherent MAG bias towards more complete assemblies for higher abundance genomes (and, in particular, for core regions of those genomes). We searched the literature for possible SAG options, including the papers referenced by the reviewer. To maximize comparability with our Tara Oceans MAGs analysis, we chose a large SAG dataset from Pachiadaki et al (2019) that sampled surface (epipelagic) ocean water in tropical and subtropical locations. These data consisted of 12,715 SAGs from planktonic bacteria and archaea, which we scanned for CRISPR systems using CRISPRCasFinder, as done previously for Tara Oceans MAGs. Approximately 17.4% of the SAGs contained at least one CRISPR array with evidence levels 1-4. However, only 0.98% of these CRISPR-containing SAGs included an evidence level 4 array, which is dramatically lower than the 9.6% observed in our MAGs dataset. We have focused our analysis exclusively on evidence level 4 arrays for a variety of reasons, as recommended by the authors of CRISPRCasFinder (Couvin et al, 2018).

Unfortunately, the evidence level 4 frequency in SAGs is too low to identify any associations with ecology ($12e3 * 0.17 * 0.01 \approx 20$ SAGs with CRISPR), even though the SAGs avoid the

coverage biases of the MAGs. We hypothesized that the low frequency of detection of CRISPR arrays in SAGs arose from the fact each single cell genome has lower coverage than a typical MAG genome. When we compared the contig length distributions between SAGs and MAGs, we found the former were significantly shorter and thus presumably less complete. Further, the repeats inherent in CRISPR arrays make them a particular challenge for assembly — and one criterion for evidence level 4 is a longer array with more observations of repeats.

Thus, while our new SAGs analysis should have had less bias, its advantage was overwhelmed by its variance and, as a result, we do not include this analysis in our revision. Instead, we have added a note to the discussion (lines 271-272) stating that “We expect these biases of MAGs will be overcome in the future with the increasing depth of coverage in studies generating large numbers of single-cell assembled genomes [59-61].”

Line 48-53: As mentioned in the general comments, this statement would be more meaningful if the authors acknowledged the impact of nutrient concentrations on the use of CRISPR Cas over surface receptor mutations (sm-mediated immunity). A key finding of reference 10 is that sm-mediated immunity is more common in high nutrient conditions, and low-nutrient conditions favor CRISPR Cas (likely due to cost of losing receptor in low nutrient environments). This would help explain discrepancies in diversity and density relationships with the prevalence of CC. This observation also ties into the results of the multi-species co-culture experiment mentioned in lines 55-59.

The reviewer raises an excellent point, and we now explicitly mention the critical role of nutrient concentration in the introduction immediately following our citation of the multi-species co-culture experiment.

As alluded to by the reviewer, while we find a relationship between CRISPR-Cas and density in ocean samples (a low nutrient environment, on average), we do not see this relationship in human-associated samples (high nutrient environment, on average). We now discuss the possible role played by nutrient density when we address these two disparate results in the Discussion on lines 274-280. The human oral environment is highly structured (e.g., with biofilms), which creates the potential for variable fine-scale nutrient concentrations. Consequently, we cannot draw definitive conclusions regarding how nutrient concentration explains discrepancies in diversity and density relationships with the prevalence of CRISPR-Cas systems.

Line 166-167: Would be meaningful about the biology of these environments if examples of prevalent genera that were removed are mentioned e.g. Pelagibacter? Prochlorococcus? Knowing these groups would provide a lot more strength to their argument that highly abundant

marine taxa almost never encode CC, as genomes from cultures of these organisms tend to lack CRISPR Cas.

We agree and have now listed the 3 most highly prevalent and abundant genera as examples in the main text (Pelagibacter, Prochlorococcus, Synechococcus; lines 165-166, 172-173) and provide prevalence and abundance for all genera in the supplementary dataset 2.

Line 254-257: The authors should also state that using MAGs fail to recover highly abundant and diverse taxa (e.g. Pelagibacter; Roda-Garcia et al 2023, Environmental Micro.), which may skew their results.

Excellent point; we have added this limitation of MAGs to the Discussion (lines 268-270). That said, it turns out that in the MAG dataset we used, a genome for Pelagibacter actually was assembled (As shown in <https://www.nature.com/articles/sdata2017203#MOESM87> supplementary table 2).

Figure 1b,d: What is "All" and "Rm" - not clearly defined in text nor figure legend. I inferred "Rm" meant rare taxa but still unclear.

We improved and clarified the meaning of "All" and "Rm" in the figure 1 caption.

Re: mSystems00568-24R1 (Ecological drivers of CRISPR immune systems)

Dear Prof. Philip L F Johnson:

Has been properly modified

Your manuscript has been accepted, and I am forwarding it to the ASM production staff for publication. Your paper will first be checked to make sure all elements meet the technical requirements. ASM staff will contact you if anything needs to be revised before copyediting and production can begin. Otherwise, you will be notified when your proofs are ready to be viewed.

Sincerely,

Francisco Rodriguez-Valera
Editor
mSystems

Reviewer #1 (Comments for the Author):

The authors addressed my concerns thoroughly and the manuscript is fit for publication. I will point out, however, that it is unlikely that the lack of CRISPR Cas detected in SAGs is simply due to variance in SAG data failing to assemble longer arrays. Short arrays are designated as level 1 evidence (Couvin et al 2018 Nucleic Acids Research) and likely do not belong to CC systems altogether (Pourcel et al. 2005 Microbiology; Couvin et al 2018 Nucleic Acids Research). The evidence levels of CRISPR arrays in CRISPRCasFinder are unrelated to Cas gene presence which would help indicate whether those level 1 arrays actually belong to a CC system. Had the authors refined what they reported based on the Cas genes and still found 17% of SAGs with cas genes but only level 1 short arrays, I would be more convinced that assembly difficulties limited the recovery of CRISPR arrays in SAGs. It is much more likely that SAGs had few CC systems simply because SAGs represent the more abundant taxa in marine samples, which tend to lack CC systems (as they demonstrate). For instance only one MAG of Pelagibacter was present out of 2631 MAGs screened (0.04%), despite Pelagibacter corresponding to 20-40% of prokaryoplankton cells per mL in the epipelagic ocean - in 3,884 SAGs of the 12,715 SAGs screened (30.5%) (Pachiadaki et al 2019 Cell). In any case, further refinement of their analysis with the presence of cas genes may have clarified some results but their findings are still valid as is.